# Multimorbidity Patterns in the Urban Population in Poland

**DOI:** 10.3390/jcm12185860

**Published:** 2023-09-08

**Authors:** Małgorzata Chlabicz, Jacek Jamiołkowski, Paweł Sowa, Magdalena Zalewska, Łukasz Kiszkiel, Mariusz Ciołkiewicz, Radosław Motkowski, Irina Kowalska, Łukasz Minarowski, Karol A. Kamiński

**Affiliations:** 1Department of Population Medicine and Lifestyle Diseases Prevention, Medical University of Bialystok, 15-269 Bialystok, Poland; mchlabicz@op.pl (M.C.); jacek909@wp.pl (J.J.); mailtosowa@gmail.com (P.S.); magdaz@umb.edu.pl (M.Z.); 2Department of Invasive Cardiology, Medical University of Bialystok, 15-276 Bialystok, Poland; 3Society and Cognition Unit, Institute of Sociology, University of Bialystok, 15-403 Bialystok, Poland; lukaszkiszkiel@gmail.com; 4Department of Rehabilitation, Medical University of Bialystok, 15-276 Bialystok, Poland; marjanc40@gmail.com; 5Department of Pediatrics, Rheumatology, Immunology and Metabolic Bone Diseases, University Children’s Hospital, Medical University of Bialystok, 15-274 Bialystok, Poland; radoslaw.motkowski@umb.edu.pl; 6Department of Internal Medicine and Metabolic Diseases, Medical University of Bialystok, 15-276 Bialystok, Poland; irina.kowalska@umb.edu.pl; 72nd Department of Lung Diseases and Tuberculosis, Medical University of Bialystok, 15-540 Bialystok, Poland; lukasz.minarowski@umb.edu.pl

**Keywords:** multimorbidity, population-based study, disability, loss of health

## Abstract

A number of studies have been conducted on multimorbidity; however, there are different patterns in various countries, ethnicities and social groups. The aim of this study is to estimate the prevalence of multimorbidity (physical diseases) in the urban population in Poland. In this population-based study, we examined multimorbidity stratified by sex, age, educational attainment and professional activity. Sixty-seven conditions were identified based on self-reported history (known conditions) and completed examinations (unknown conditions). Among the overall individuals aged 20–80 years, 1422 (88.2%) of the total 1612 individuals, 787 (88.9%) of 885 women and 635 (87.3%) of 727 men were diagnosed with at least two chronic conditions. On average, 5.25 ± 3.5 conditions occurred in the study population. The number of diagnosed conditions per individual increased with age and decreased with higher educational levels, with differing pathways in women and men. Women showed a higher number of conditions than men in the same age groups and educational levels. Only among students, the level of multimorbidity was lower in women than in men. In the other occupational activity categories, it was already higher in women. The level of multimorbidity in employed and unemployed individuals in a particular sex cluster was similar. We identified a high prevalence of multimorbidity in the urban population in Poland varying by age, sex, education attainment and professional activity. Our work may help in the selection of appropriate screening tests based on age, sex and educational attainment in order to recognise multimorbidity based on both known and unknown conditions. Ultimately, it may impact clinical practice, service delivery and study design.

## 1. Introduction 

Multimorbidity, defined as the coexistence of two or more chronic conditions in the same individual, is significant for research as well as medical practice [1,2]. Multimorbidity has become a major concern of public health, health care systems and health care providers [3]. Multimorbidity has increased due to an ageing population, changing lifestyles, better socio-economic conditions and better treatment options [2]. The incidence of multimorbidity depends on sex, age or educational attainment [1]. The prevalence of multimorbidity in the urban population is higher than in the rural population [4]. People with multimorbidity have higher mortality rates [5], high health care utilisation and an increase in expenses [6,7]. Identifying these diseases, their prevalence and common combinations can help to design studies or ensure an adequate health service supply. Moreover, previous studies tended to use hospital data or data from medical databases [1] or include only the older population [3], which may have overlooked common conditions in the general population that were not yet diagnosed. Part of the population is unaware that they have a disease and therefore do not declare these diseases during the self-reporting history, and they do not exist in medical databases. Another problem is that most studies exclude young people who are potentially healthy (not yet in contact with the health service), despite evidence that multimorbidity can also affect younger people, and patterns are likely to be different from those in older age groups. In addition, few studies to date have considered the dependence on professional activity, which may result in different patterns of multimorbidity. In our study, we addressed this issue. We assessed multimorbidity from the general population (with a young people) and assessed multiple parameters to identify diseases in those unaware of them by providing detailed research at the research centre.

The severity of a disability is an important factor in estimating the length of healthy life loss due to a particular condition [8]. Thus, several sets of scales for the disabled have been developed. One of these is the Global Burden of Disease Study 2019 (GBD 2019) Disability Scales [9,10], which estimated the burden of diseases, injuries and risk factors for 204 countries and territories.

The aim of this study was to estimate the prevalence of multimorbidity (physical diseases) in the general urban population in Poland stratified by sex, age, educational level and professional activity. An additional objective was to use the Global Burden of Disease Study 2019 (GBD 2019) Disability Scale to assess the loss of health by disability weight.

## 2. Materials and Methods

The Bialystok PLUS is a single-centre cohort study. The survey has been conducted since 2018 on a sample of Bialystok (Poland) residents aged 20–80 years. Participants were selected at random from among the residents of Bialystok to achieve a distribution of proportions similar to the city’s population. Each year, we received a pseudonymised list of Bialystok citizens from the Municipal Office in Bialystok. The dataset was limited to people aged 20–79 years old, and categories were based on sex and 5-year intervals. We randomly sampled citizens from each subcategory separately, in such a number that allowed us to obtain a similar proportion distribution similar to that in the city’s population [11]. There were not any exclusions; there were only some restrictions. Participants with an acute infectious disease or who had surgery within the last six weeks were not examined and they were encouraged to come back to the study after this period. Moreover, during the coronavirus disease 2019 (COVID-19) pandemic, reverse transcription polymerase chain reaction (RT-PCR) was performed with nasopharyngeal swabs using the CFX96 Real-Time System (Bio-Rad, Hercules, CA, USA) to exclude active COVID-19 infection. At the time of the study, there were exclusions for individual procedures, e.g., pregnancy for dual-energy X-ray absorptiometry (DEXA) and diabetes for oral glucose tolerance test (OGTT).

The high replicability of the tests was obtained by conducting them according to validated Standard Operating Procedures (SOPs). At the time of entry into the study, demographic characteristics, information on participants’ medical history, education and work activity were collected from questionnaires. In addition, a comprehensive assessment was carried out, allowing for many chronic diseases to be recognised in those unaware of them. A list of parameters with the method and equipment used is presented in Table 1.

Sixty-seven chronic conditions were identified according to self-reported questionnaire. Seventeen of the declared entities were confirmed or newly diagnosed in people unaware of their presence. The Mini Mental State Examination (MMSE) is a 30-item questionnaire used to measure cognitive impairment that was completed for participants over 50 years of age. Dementia was diagnosed when the MMSE scale score was below 24 points in the population with primary and secondary education and below 26 points in the population with higher education. Undiagnosed diabetes was defined as fasting glucose ≥ 126 mg/dL or OGGT 120 min glucose ≥ 200 mg/dL or HbA1c ≥ 6.5% in probands with no history of diabetes. Undiagnosed hypercholesterolemia was identified when the proband had no history of hypercholesterolemia, but had TC > 190 mg%. Undiagnosed triglyceridemia was established when TG > 150 mg% was found in the proband with a history of triglyceridemia. Undiagnosed kidney disease was identified when the proband had no history of kidney disease, but had eGFR < 60 mL/min/1.73 m^2^. Hyperthyroidism was defined at TSH ≤ 0.004 µIU/mL, and hypothyroidism was defined at TSH > 4.2 µIU/mL with normal anti-TPO and anti-TG values. Hashimoto’s disease was identified when anti-TG levels were over 115 IU/mL or/and when anti-TPO was over 34 IU/mL. Undiagnosed hypertension was identified when the proband without a history of hypertension had BPs ≥ 140 and/or BPd ≥ 90 mmHg. At a BMI of 30.0 or higher, obesity was diagnosed. Undiagnosed peripheral artery disease (PAD) was identified when the proband had no history of PAD, but had ABI ≤ 0.9 measured on at least one side. Undiagnosed heart failure (HF) was identified when the proband had no history of HF, but had LVEF < 50%. Atrial fibrillation/flutter (AF/AFl) was identified when the ECG showed atrial arrhythmia. FEV1/FVC < 0.7 (70%) detected airflow obstruction in chronic obstructive pulmonary disease (COPD) [12]. Osteoporosis was defined in DEXA when T scores were ≤−2.5 in women ≤ 50 years and in men under 65 years, and Z scores ≤ −2.5 in the remaining participants. Goiter refers to enlargement of the thyroid gland over 20 mL in women and over 25 mL in men. A focal thyroid lesion was identified if at least one of the three measurements of the lesion was above 10 mm.

For the final assessment of multimorbidity, self-reported chronic conditions (known morbidity) and newly diagnosed diseases (unknown morbidity) during the study were taken into account (Appendix A). Multimorbidity was defined as two or more chronic conditions occurring simultaneously in the same person.

Furthermore, we used the Global Burden of Disease Study 2019 (GBD 2019) Disability Scales [8,9] to assess the severity of disability due to a particular condition. A disability weight is a weighting factor that reflects the relative severity of a health state, with a value anchored from 0 to 1, with 0 implying a state that is equivalent to full health, and 1 implying a state equivalent to death. Thirty-seven of the sixty-seven chronic conditions in our survey were assessed according to the GBD 2019 Disability Scales. Due to the lack of data on the severity of a particular disease in our study, an average value was calculated from all disability weight relating to a particular disease and then used for further analysis for the current study, e.g., idiopathic, severe epilepsy: disability weight 0.552; idiopathic, less severe epilepsy: disability weight 0.263; idiopathic, seizure-free, treated epilepsy: disability weight 0.049; epilepsy: mean value of disability weight 0.288.

## 3. Ethics Statements

Ethical approval for Bialystok PLUS study was provided by the Ethics Committee of the Medical University of Bialystok (Poland) on 31 March 2016 (approval number: R-I-002/108/2016). Informed consent was obtained from all subjects involved in the study. The study was conducted in accordance with the Declaration of Helsinki.

## 4. Statistical Analysis

The statistical analysis was performed using the IBM SPSS Statistics 27.0 software (SPSS, Armonk, NY, USA). Descriptive statistics were calculated for sex, age, educational attainment and professional activity by the number of chronic conditions. Descriptive statistics for quantitative variables were presented as means and standard deviations (SD). Categorical variables are presented as frequencies (n) and percentages (%). The Pearson’s χ^2^ tests were used for comparison between the groups for categorical variables. The statistical significance was considered when *p* ≤ 0.05. The figures were prepared using the R version 4.2.2 software (www.R-project.org accessed on 28 July 2023) and the ggplot2 library.

## 5. Results

We derived multimorbidity patterns from 1612 individuals, of whom 885 (54.9%) were female, 727 (45.1%) were male, 42 (2.6%) received primary education, 747 (46.4%) received secondary education and 820 (51.0%) received higher education. A total of 27 (1.7%) individuals were students, 1101 (68.9%) were employed, 86 (5.4%) were unemployed and 384 (24.0%) were pensioners (Table 2).

Among the overall individuals aged 20–80 years, 1422 (88.2%) of the total 1612; 787 (88.9%) of 885 women and 635 (87.3%) of 727 of men had been diagnosed with at least two conditions. The number of diagnosed conditions per individual increased with age, with differing pathways in women and men. Women showed a higher number of conditions than men in the same age groups (Table 3 and Table 4; Figure 1).

The prevalence of multimorbidity increased with age and decreased with a higher educational level, with differing pathways in women and men. Women showed a higher number of conditions than men in the same age groups or educational levels. The lowest recorded multimorbidity levels (4.2 ± 3.1) were observed in men with higher education (Table 3 and Table 4).

Among students, the level of multimorbidity was lower in women (1.3 ± 0.9) than in men (1.6 ± 1.2). In the other occupational activity categories, it was already higher in women. The level of multimorbidity in employed and unemployed individuals in a particular sex cluster was similar (Table 3 and Table 4).

The most common disease entities in the general population were hypercholesterolemia (59.9%), obesity (46.9%), hypertension (40.3%) and focal thyroid lesions (30.6%). Detailed data are included in the Appendix A and are shown in Figure 2.

The prevalence of particular diseases varied according to sex, educational level and professional activity.

The most common diseases in the male population were hypercholesterolemia (61.5%), hypertension (50.2%) obesity (48.1%) and hypertriglyceridemia (37.3%), whereas the most common conditions in the female population were hypercholesterolemia (58.6%), obesity (45.9%), focal thyroid lesions (38.9%) and osteoarthritis (37.9%). The prevalence of multiple conditions differed statistically significantly by sex. Detailed data are included in the Appendix A and are shown in Figure 3.

The most common diseases in the primary education subgroup were hypercholesterolemia (66.7%), obesity (64.3%), hypertension (54.8%) and hypertriglyceridemia (47.6%). The most common conditions in the secondary education subgroup were hypercholesterolemia (64.5%), obesity (49.4%), hypertension (48.3%) and osteoarthritis (39.6%), whereas the most common conditions in the higher education subgroup were hypercholesterolemia (55.5%), obesity (43.8%), hypertension (32.3%) and focal thyroid lesions (28.8%). The prevalence of a few entities differed statistically significantly in terms of educational level. Detailed data are included in the Appendix A and are shown in Figure 4.

The most common diseases in the subgroup of students were allergy (29.6%), obesity (22.2%), gastroesophageal reflux disease (22.2%) and hypertension (14.8%). The most common conditions in the employed subgroup were hypercholesterolemia (56.6%), obesity (44.1%), hypertension (33.2%) and hypertriglyceridemia (26.4%), whereas the most common conditions in the unemployed subgroup were hypercholesterolemia (58.1%), obesity (44.2%), hypertension (31.4%) and osteoarthritis (28.8%). The most common diseases in the subgroup of pensioners were hypercholesterolemia (74.0%), hypertension (65.4%), obesity (57.0%) and osteoarthritis (55.5%). Detailed data are included in the Appendix A and are shown in Figure 5.

In addition, we assessed the health burden of the identified diseases in our study using the Global Burden of Disease Study 2019 (GBD 2019) Disability Scales [10]. In Table 5 and Appendix A, we provide a detailed calculation of the mean loss of health in the overall population by disability weight according to the Global Burden of Disease Study 2019 (GBD 2019) Disability Scales.

Based on the patient-declared and newly diagnosed diseases, the average health loss among the urban population was 0.238. Migraine (12.6%), osteoarthritis (9.13%) and discopathy (8.99%) were the most significant factors in the total health loss. The average health loss values were 0.197 for men and 0.272 for women. The most significant relative role in the total health loss among men was established to be diabetes (11.6%), and among women, migraine (17.3%) was the most significant. In the subgroup with a primary education, the average health loss was estimated at 0.361; in the subgroup with a secondary education, it was estimated at 0.279 and in the subgroup with a higher education, it was estimated at 0.194. The most important relative role in the total health loss among those with a primary education was migraine (12.8%), among those with a secondary education, the most important relative role was osteoarthritis (10.1%) and among those with a higher education, the most important relative role was migraine (16.8%). In the students, the average health loss was estimated at 0.055; in the working individuals, it was estimated at 0.183; in the unemployed, it was estimated at 0.207 and in the pensioners, it was estimated at 0.491. On the other hand, the most significant relative role in the total health loss in students was gastroesophageal reflux disease (29.6%), in the employed and the unemployed, the most significant relative role was migraine (16.8% and 15,5%, respectively), and in the pensioner subgroup, the most significant relative role was diabetes (10.6%). Detailed data are included in the Appendix A.

## 6. Discussion

Using the data from 1612 randomly selected participants aged 20–80 years old, we identified differences in multimorbidity by age, sex, education attainment and professional activity. The prevalence of multimorbidity increased with age and decreased with higher educational levels. Women showed a higher number of conditions than men in the same age groups and educational levels.

A comparison with previous reports is difficult because of the differences in the sample characteristics [3,13]. A high heterogeneity between studies in the meta-analysis implies that the prevalence of multimorbidity varies between studies [2]. Chowdhury et al. [2] analysed data from 126 peer-reviewed studies and showed that the prevalence of multimorbidity among the adult population ranged from 4.0% to 92.8%. The random-effects overall pooled estimated prevalence of multimorbidity was 37.2%. Souza et al. [3], using eleven non-communicable conditions included in the definition of multimorbidity, showed that the multimorbidity rate in Poland in the 2004–2017 period was about 45% in people aged 50 years and over. Compared with this analysis, we found a higher prevalence of multimorbidity, whereas Piotrowicz et al. [14] in a nationwide, multicentre, cross-sectional PolSenior study of patients aged 55 years and older showed that multimorbidity was identified in 70.9% of participants aged 55 to 59 years, with 88.4% of those aged 65 to 80 years and 93.9% of those aged 80 years and older. Furthermore, the prevalence of multimorbidity increased with the number of conditions included in the definition of multimorbidity. Kuan et al. [1] showed in an English-population-based study that the prevalence of multimorbidity in the population of people who are 30–39 years old was 65.8%, 77.4% for those who are 40–49 years old, 86% for those who are 50–59 years old, 93% for those who are 60–69 years old and 97% for those who are 70–80 years old using 308 selected physical and mental health conditions. In our study, diseases were identified not only via interviews, but also via multidirectional testing, as many people are unaware of the presence of these diseases. In the current study, 50% of diabetics, 29% of hypertensive patients and 44.7% of patients with hypercholesterolemia were unaware that they had these diseases. Secondly, in our study, the number of conditions included to define multimorbidity was significant at sixty-seven chronic conditions, which may explain such a high multimorbidity prevalence in our analysis.

Previous data show that the prevalence of multimorbidity varies by gender and educational attainment. Chowdhury et al. [2] also showed that multimorbidity was more common in women (39.4%) than in men (32.8%). Li et al. [15] showed that females were 1.7 times more likely to experience multimorbidity than males (odds ratio (OR) 1.72, 95% CI 1.69–1.74). Our research showed that women had a higher number of medical conditions than men in the same age groups or at the same levels of education, whereas the prevalence of multimorbidity occurred with a similar frequency in both sexes. Frølich et al. [16] and Tan et al. [17] showed that the prevalence of multimorbidity was inversely related to educational attainment, which we confirmed in our study.

The pattern of the most common diseases varies substantially according to sex, age, educational level and work activity. Overall, the most common diseases were hypercholesterolaemia, hypertension, obesity, hypertriglyceridemia, focal thyroid lesions and osteoarthritis. Not only did the order differ in the clusters analysed, but also the prevalence. Individuals with a low education level and pensioners had the highest prevalence of diseases. Souza et al. [3] showed that cardiometabolic and musculoskeletal diseases were more prevalent, while cancer and neurodegenerative diseases were less prevalent. Hypertension and hyperlipidaemia was the most common combination found in 30.8 million American adults in a study by Schiltz et al. [18], whereas hypertension, hyperlipidaemia and diabetes was the most common three-way combination, and hypertension, hyperlipidaemia, musculoskeletal disorder, diabetes and arthritis was the most common five-way combination. Combinations including hypertension and/or hyperlipidaemia plus another condition comprise the majority of the most prevalent combinations [18]. Sierpiński et al. [19], in an article describing comorbidities coexisting with HF in the Polish population, showed that hypertension and coronary artery disease were the most prevalent comorbidities in the cohort of Polish HF patients. Piotrowicz et al. [14], in a PolSenior study, showed that hypertension constituted the most frequent single diagnosis. They also showed that metabolic disease, obesity, arrhythmia, mental illness and thyroid disease were conditions that each occurred more commonly in than 1% of participants under 80 years of age, while cognitive impairment was more common in the octogenarian and older groups [14]. Similar conditions prevailed in our study.

We used the Global Burden of Disease Study 2019 (GBD 2019) Disability Scales [9,10] to assess the severity of disability due to a particular condition. Disability weight, a parameter for calculating disease burden, is a weight factor that reflects the severity of health state from disease or injury. In our study, the average health loss was higher among women, decreased with a higher educational level and increased with age, with different disease patterns in their relative roles in the total health loss. Schiltz et al. [18] showed that the combination of diabetes–arthritis–asthma/COPD had the highest age/race/sex adjusted odds ratio of poor self-rated health. Hilderink et al. [20] analysed 25 health conditions from the Dutch Burden of Disease study that have high rates of prevalence and contribute to the total number of Years Lived with a Disability (YLD). They showed that calculations of the burden of disease that do not take account multimorbidity can result in an overestimation of the true burden of disease. This may have implications for public health policy strategies that focus on single conditions.

We identified a high prevalence of multimorbidity in an urban population in Poland based on known and unknown conditions. This may be highly relevant for health care providers and may focus the search for unknown disease entities, and hence, multimorbidity in the relevant subpopulations. Our work has shown that more frequently, unrecognised diseases were more common in the subgroup with a primary education. The early detection and treatment of these diseases can result in lower health care costs, since a significant increase in the use of health care was shown for people with multimorbidity compared to people without chronic diseases [16].

Our data might be useful for health care service providers to select appropriate care procedures according to sex, age, education attainment and professional activity, which may impact clinical practice, service delivery and research design.

## 7. Study Limitation

This study has some limitations. It is a single-centre study with a limited sample size of urban residents. In our study, we did not assess mental health, and therefore, the Charlson index [21] could not be used for our analysis. Not all disease entities declared in the interview could be confirmed or newly discovered during the examination at the research centre. Nevertheless, this study has the advantage of analysing a general population of randomly selected individuals including a group of young people (low users of health care). Another major advantage is that all of these people were thoroughly examined, allowing for the detection of many diseases of which the individuals were unaware of having.

## 8. Conclusions

The high prevalence of multimorbidity highlights the need for health care adjustments to this increasing problem. Our work may help in the selection of appropriate screening tests based on age, sex and educational attainment in order to recognise both known and unknown conditions. Ultimately, it may have an impact on clinical practice, service delivery and study design.

## Figures and Tables

**Figure 1 jcm-12-05860-f001:**
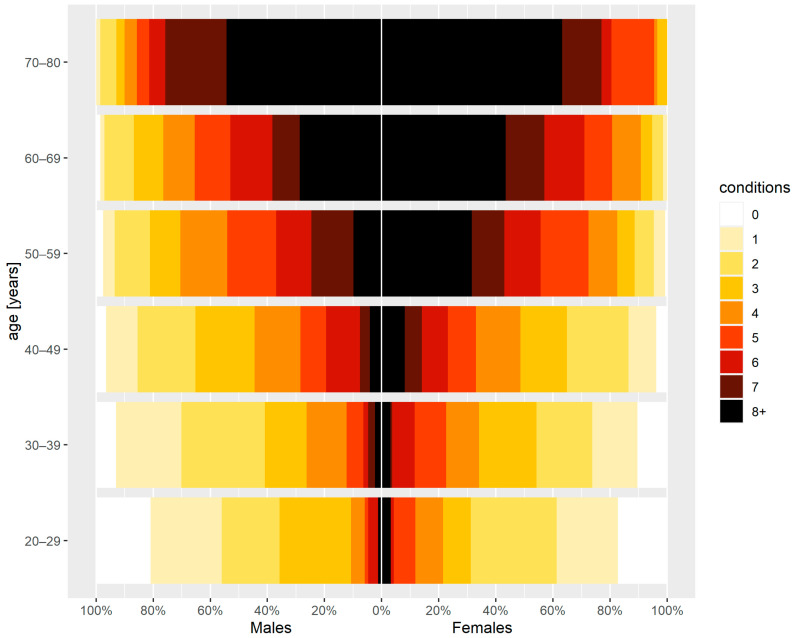
Number of conditions per individual stratified by sex and age.

**Figure 2 jcm-12-05860-f002:**
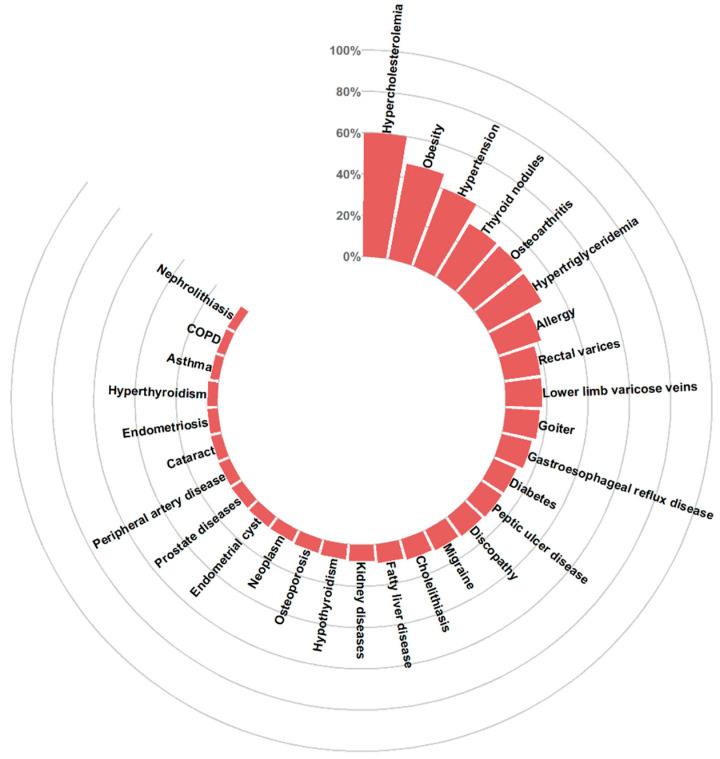
Multimorbidity of the top 30 conditions in the general urban population.

**Figure 3 jcm-12-05860-f003:**
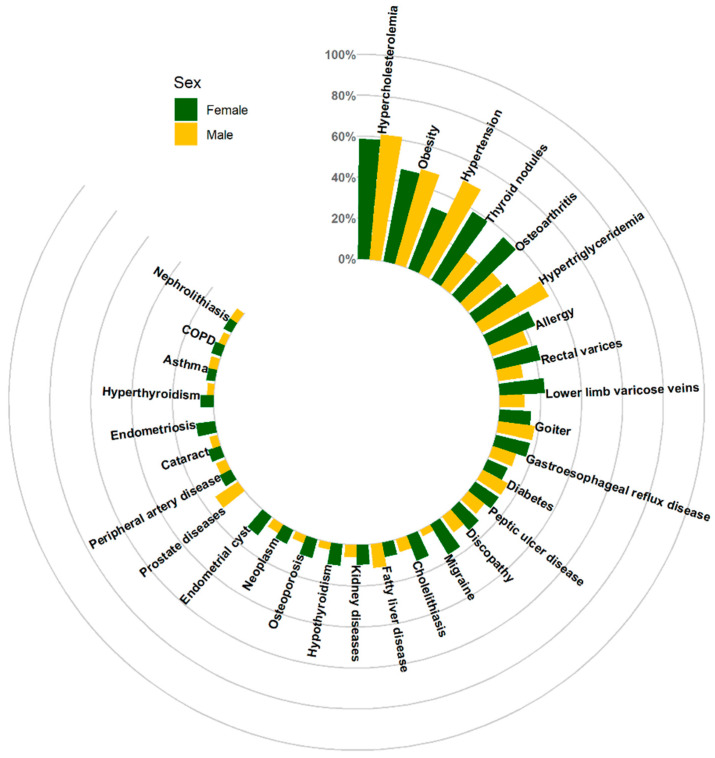
Multimorbidity of the top 30 conditions stratified by sex.

**Figure 4 jcm-12-05860-f004:**
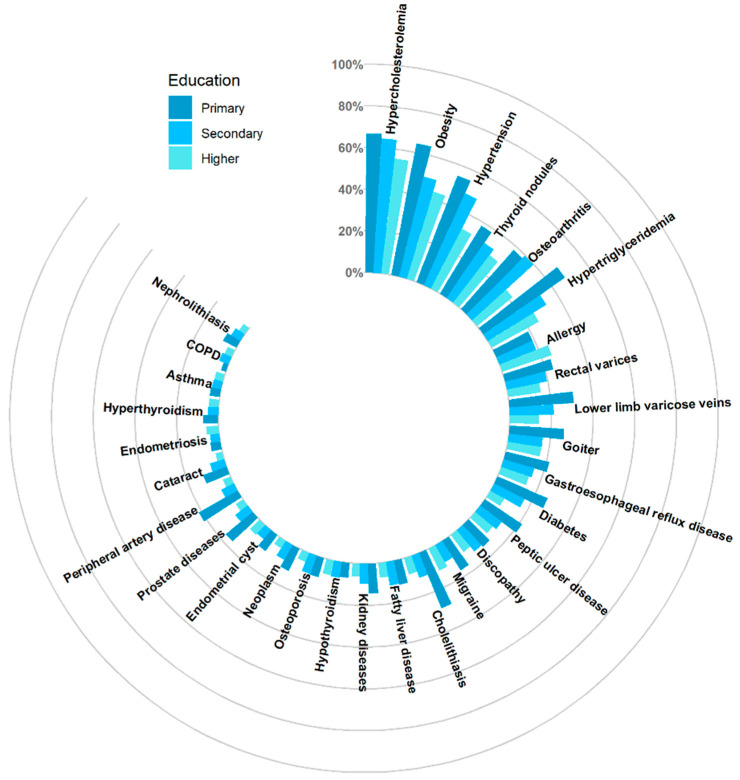
Multimorbidity of the 30 most common conditions according to educational level.

**Figure 5 jcm-12-05860-f005:**
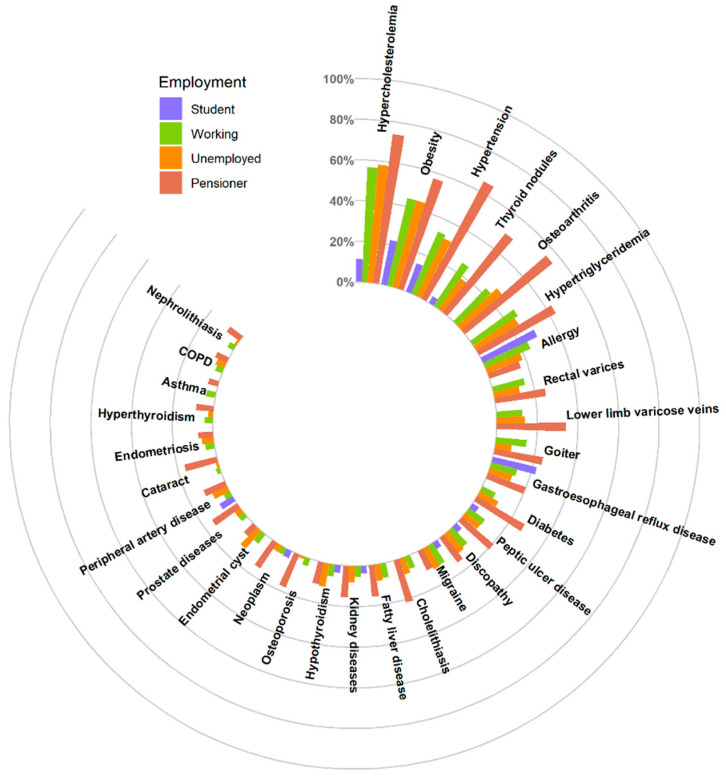
Multimorbidity of the 30 most common conditions according to professional activity.

**Table 1 jcm-12-05860-t001:** Parameters, methods and equipment used in the study.

Parameter	Method	Device
Fasting glucose and the 120 min glucose in oral glucose tolerance test (OGTT)	hexokinase method	Cobas e111, ROCHE Diagnostic Ltd., Rotkreuz, Switzerland
Haemoglobin A1c (HbA1c)	ion-exchange high-performance liquid chromatography (HPLC)	Bio-Rad, Hercules, CA, USA
Total cholesterol (TC)	enzymatic colorimetric method	Cobas e111, ROCHE Diagnostic Ltd., Rotkreuz, Switzerland
Triglycerides (TG)	enzymatic colorimetric method	Cobas e111, ROCHE Diagnostic Ltd., Rotkreuz, Switzerland
Creatinine	enzymatic colorimetric method	Cobas e111, ROCHE Diagnostic Ltd., Rotkreuz, Switzerland
Serum thyroid-stimulating hormone (TSH)	electrochemiluminescence method (ECLIA)	Cobas e411, ROCHE Diagnostic Ltd., Rotkreuz, Switzerland
Triiodothyronine (fT3)	electrochemiluminescence method (ECLIA)	Cobas e411, ROCHE Diagnostic Ltd., Rotkreuz, Switzerland
Thyroxine (fT4)	electrochemiluminescence method (ECLIA)	Cobas e411, ROCHE Diagnostic Ltd., Rotkreuz, Switzerland
Antithyroid peroxidase (anti-TPO) antibodies	electrochemiluminescence method (ECLIA)	Cobas e411, ROCHE Diagnostic Ltd., Rotkreuz, Switzerland
Anti-thyroglobulin (anti-Tg)	electrochemiluminescence method (ECLIA)	Cobas e411, ROCHE Diagnostic Ltd., Rotkreuz, Switzerland
Blood pressure (BP)	oscillometric method	Omron Healthcare Co. Ltd. MG Comfort device
Ankle-brachial index (ABI)	oscillometric method	Vascular Explorer, Enverdis, Jena, Germany
Echocardiography	ultrasound method	GE Healthcare, Chicago, IL, USA
Thyroid gland	ultrasound method	GE Healthcare, Chicago, IL, USA
Resting electrocardiography (ECG)	electrocardiography method	AMEDTEC Medizintechnik Aue GmbH, Aue, Germany
Spirometry	pulmonary functional method	BodyBox 5500 cabin, Warsaw, Poland
Bone mass	dual-energy X-ray absorptiometry (DEXA)	GE Healthcare, Chicago, IL, USA

**Table 2 jcm-12-05860-t002:** Characteristics of the study population.

	Population N = 1612	Number of Diagnosed Conditions	Number of Individuals with Two or More Diagnosed Conditions	*p* *
Overall	1612	5.25 ± 3.5	1422 (88.2%)	-
Sex				
Female	885 (54.9%)	5.7 ± 3.6	787 (88.9%)	0.327
Male	727 (45.1%)	4.7 ± 3.3	635 (87.3%)	
Age group (years)				
20–29	177 (11.0%)	2.42 ± 1.9	104 (58.8%)	<0.001
30–39	335 (20.8%)	3.13 ± 2.1	241 (71.9%)	
40–49	329 (20.4%)	4.1 ± 2.3	283 (86.0%)	
50–59	271 (16.8%)	6.1 ± 2.8	256 (94.5%)	
60–69	343 (21.3%)	7.4 ± 3.4	336 (98.0%)	
70–80	157 (9.7%)	9.3 ± 3.7	156 (99.4%)	
Education				
Primary	42 (2.6%)	7.7 ± 4.7	41 (97.6%)	<0.001
Secondary	747 (46.4%)	5.8 ± 3.5	684 (91.6%)	
Higher	820 (51.0%)	4.6 ± 3.3	695 (84.8%)	
Professional activity				
Student	27 (1.7%)	1.4 ± 1.1	13 (48.1%)	<0.001
Working	1101 (68.9%)	4.4 ± 2.9	942 (85.6%)	
Unemployed	86 (5.4%)	4.4 ± 2.6	75 (87.2%)	
Pensioner	384 (24.0%)	8.3 ± 3.6	380 (99.0%)	

Data are shown as n (%) or mean ± SD. * The statistical analysis refers to the number of people with two or more diagnosed conditions.

**Table 3 jcm-12-05860-t003:** Characteristics of the study female population.

	Population N = 885	Number of Diagnosed Conditions	Number of Individuals with Two or More Diagnosed Conditions
Age group (years)			
20–29	93 (10.5%)	2.5 ± 2.1	61 (65.6%)
30–39	164 (18.5%)	3.3 ± 2.2	125 (76.2%)
40–49	185 (20.9%)	4.1 ± 2.4	163 (88.1%)
50–59	149 (16.8%)	6.8 ± 3.0	146 (98.0%)
60–69	207 (23.4%)	7.9 ± 3.3	205 (99.0%)
70–80	87 (9.8%)	9.7 ± 3.5	87 (100.0%)
Education			
Primary	15 (1.7%)	9.9 ± 4.5	15 (100.0%)
Secondary	390 (44.2%)	6.5 ± 3.6	362 (92.8%)
Higher	478 (54.1%)	4.9 ± 3.3	409 (85.6%)
Professional activity			
Student	14 (1.6%)	1.3 ± 0.9	5 (35.7%)
Working	550 (63.8%)	4.6 ± 3.0	467 (83.4%)
Unemployed	50 (5.8%)	4.5 ± 2.8	43 (86.0%)
Pensioner	248 (28.8%)	8.6 ± 3.5	246 (99.2%)

Data are shown as n (%) or mean ± SD.

**Table 4 jcm-12-05860-t004:** Characteristics of the study male population.

	Population N = 727	Number of Diagnosed Conditions	Number of Individuals with Two or More Diagnosed Conditions
Age group (years)			
20–29	84 (11.6%)	2.3 ± 1.8	55 (65.5%)
30–39	171 (23.5%)	3.0 ± 2.0	131 (76.6%)
40–49	144 (19.8%)	4.1 ± 2.3	131 (91.0%)
50–59	122 (16.8%)	5.3 ± 2.4	116 (95.1%)
60–69	136 (18.7%)	6.7 ± 3.4	133 (97.8%)
70–80	70 (9.6%)	8.8 ± 3.9	69 (98.6%)
Education			
Primary	27 (3.7%)	6.4 ± 4.4	26 (96.3%)
Secondary	357 (49.2%)	5.1 ± 3.3	322 (90.2%)
Higher	342 (47.1%)	4.2 ± 3.1	286 (83.6%)
Professional activity			
Student	13 (1.9%)	1.6 ± 1.2	6 (46.2%)
Working	541 (74.5%)	4.1 ± 2.7	436 (80.6%)
Unemployed	36 (5.0%)	4.2 ± 2.5	28 (77.8%)
Pensioner	136 (18.6%)	7.8 ± 3.7	134 (98.5%)

Data are shown as n (%) or mean ± SD.

**Table 5 jcm-12-05860-t005:** Mean loss of health and relative role in total health loss by disability weight according to the Global Burden of Disease Study 2019 (GBD 2019) Disability Scales.

Disease	Mean Loss of Health by Disability Weight in Overall Population	Relative Role in Total Health Loss in Overall Population
Overall	0.238	-
Migraine	0.030	12.60%
Osteoarthritis	0.022	9.13%
Discopathy	0.021	8.99%
Kidney diseases	0.020	8.38%
Diabetes	0.020	8.28%
Chronic obstructive pulmonary disease (COPD)	0.019	8.10%
Fatty liver disease	0.011	4.62%
Gastroesophageal reflux disease	0.011	4.58%
Psoriasis	0.010	4.33%
Gout	0.007	3.05%
Hyperthyroidism	0.007	3.02%
Cataract	0.007	3.00%
Nephrolithiasis	0.006	2.69%
Rheumatoid arthritis	0.006	2.45%
Endometriosis	0.006	2.40%
Atrial fibrillation	0.005	1.93%
Heart failure	0.004	1.70%
Neoplasm	0.003	1.44%
Alzheimer’s and other dementias	0.003	1.32%
Stroke	0.003	1.19%
Glaucoma	0.003	1.19%
Asthma	0.002	0.84%
Goiter	0.002	0.77%
Hypothyroidism	0.001	0.59%
Age-related macular degeneration	0.001	0.56%
Epilepsy	0.001	0.53%
Parkinson’s disease	0.001	0.44%
Ulcerative colitis	0.001	0.43%
Hepatitis C	0.001	0.35%
Cirrhosis of the liver	0.001	0.22%
Hepatitis B	0.001	0.22%
Myocarditis	0.001	0.21%
Chronic pancreatitis	0.0004	0.18%
Peripheral artery disease	0.0004	0.17%
Polycystic ovary syndrome (PCOS)	0.0001	0.06%
Human immunodeficiency virus (HIV)	0.0001	0.05%
Myocardial infarction	0.00	0.00%

## Data Availability

The datasets are not publicly available because the individual privacy of the participants should be protected. Data are available from the corresponding authors upon reasonable request.

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
