# Peer review of "Multimorbidity Patterns in the Urban Population in Poland"

_jcm, 2023, doi:10.3390/jcm12185860_

Round 1
Reviewer 1 Report
I reviewed with interest the manuscript of Chlabicz et al. "Multimorbidity patterns in the urban population". In this article, the authors examined in a population-based study multimorbidity stratified by sex, age, educational attainment and professional activity. A feature of the study was the examination of the entire cohort of residents aged 20-79 years, as well as an additional examination aimed at identifying previously undiagnosed diseases. As a result, some new scientific facts have been obtained that can help policymakers maximize existing resources to reduce the burden and adverse effects of multimorbidity.
At the same time, when reviewing, I had questions and comments to which I would like to receive answers from the authors.
1. It is noteworthy that the authors interpret the concept of "multimorbidity" very broadly, including not only known diseases, but also risk factors (such as hypercholesterolemia, hypertriglyceridemia, etc.). Not surprisingly, the average number of diseases/conditions among the authors was 5.25. In my opinion, such an extended interpretation of the term "multimorbidity" confuses practitioners. Since the treatment of diseases and the fight against their risk factors require various organizational and therapeutic measures. Therefore, these concepts should not be confused.
2. The authors rightly note that the prevalence of "multimorbidity" depends on the set of diseases considered in the study. Therefore, the data of different authors is very difficult to compare with each other. Why did the authors not use, for example, the well-known Charlson index, the use of which would allow a wider dissemination of the results of this study? It is also advisable for the authors to justify why they chose such a set of indicators for assessing "multimorbidity" (in the introduction or in the discussion)
3. The topic of multimorbidity attracts the attention of many researchers, in my opinion, in the introduction and discussion, it is advisable for the authors to consider the following recent publications (see 1-3).
4. The authors refer to their 2022 publication (ref. 10) when describing the inclusion of participants in the study. At the same time, the recruitment is also not fully described in that article; a link is given to another study by the authors (4). This is not convenient for readers, the authors should correct this section of the article.
References:
1. Li XL, Huang H, Lu Y, Stafford RS, Lima SM, Mota C, Shi X. Prediction of Multimorbidity in Brazil: Latest Fifth of a Century Population Study. JMIR Public Health Surveill. 2023 May 30;9:e44647. doi: 10.2196/44647.
2. Tan MMC, Prina AM, Muniz-Terrera G, Mohan D, Ismail R, Assefa E, Keinert AÁM, Kassim Z, Allotey P, Reidpath D, Su TT. Prevalence of and factors associated with multimorbidity among 18 101 adults in the South East Asia Community Observatory Health and Demographic Surveillance System in Malaysia: a population-based, cross-sectional study of the MUTUAL consortium. BMJ Open. 2022 Dec 23;12(12):e068172. doi: 10.1136/bmjopen-2022-068172.
3. Singh K, Alomari A, Lenjawi B. Prevalence of Multimorbidity in the Middle East: A Systematic Review of Observational Studies. Int J Environ Res Public Health. 2022 Dec 8;19(24):16502. doi: 10.3390/ijerph192416502.
4. Chlabicz M, Jamiołkowski J, Paniczko M, Sowa P, Szpakowicz M, Łapińska M, Jurczuk N, Kondraciuk M, Ptaszyńska-Kopczyńska K, Raczkowski A, Szpakowicz A, Kamiński KA. ECG Indices Poorly Predict Left Ventricular Hypertrophy and Are Applicable Only in Individuals With Low Cardiovascular Risk. J Clin Med. 2020 May 6;9(5):1364. doi: 10.3390/jcm9051364.
No comments
Author Response
Reviewer #1
I reviewed with interest the manuscript of Chlabicz et al. "Multimorbidity patterns in the urban population". In this article, the authors examined in a population-based study multimorbidity stratified by sex, age, educational attainment and professional activity. A feature of the study was the examination of the entire cohort of residents aged 20-79 years, as well as an additional examination aimed at identifying previously undiagnosed diseases. As a result, some new scientific facts have been obtained that can help policymakers maximize existing resources to reduce the burden and adverse effects of multimorbidity.
At the same time, when reviewing, I had questions and comments to which I would like to receive answers from the authors.
- It is noteworthy that the authors interpret the concept of "multimorbidity" very broadly, including not only known diseases, but also risk factors (such as hypercholesterolemia, hypertriglyceridemia, etc.). Not surprisingly, the average number of diseases/conditions among the authors was 5.25. In my opinion, such an extended interpretation of the term "multimorbidity" confuses practitioners. Since the treatment of diseases and the fight against their risk factors require various organizational and therapeutic measures. Therefore, these concepts should not be confused.
Thank you for your comments. We agree, there is a problem with the identification of disease entities included in the term multimorbidity. We thought about the choice for a long time. After reviewing numerous publications that identified lipid disorders as one of the disease entities of multimorbidity [1-5]and from clinical experience (these are serious disorders requiring pharmacological treatment, not just lifestyle modification), we decided to include lipid disorders in our analysis.
- The authors rightly note that the prevalence of "multimorbidity" depends on the set of diseases considered in the study. Therefore, the data of different authors is very difficult to compare with each other. Why did the authors not use, for example, the well-known Charlson index, the use of which would allow a wider dissemination of the results of this study? It is also advisable for the authors to justify why they chose such a set of indicators for assessing "multimorbidity" (in the introduction or in the discussion)
We did not use the Charlson index[6] in our analysis due to the lack of data on mental illness - as we mentioned in the Study limitations. In the section Study limitation, we have added: 'In our study we did not assess mental health, and therefore the Charlson index could not be used for our analysis.’
- The topic of multimorbidity attracts the attention of many researchers, in my opinion, in the introduction and discussion, it is advisable for the authors to consider the following recent publications (see 1-3).
Thank you for pointing out interesting publications. Li et al. analyzed the impact of demographic factors and predicted the impact of various risk factors on multimorbidity in Portugal. Tan et al. evaluated multiculturalism in Malaysia. The study shows that multimorbidity is common among adults of any age. The prevalence and risk of multimorbidity increased with age; is associated with sex, education level and employment status. Singh et al. measured the prevalence, demographic factors and consequences of multimorbidity in the Middle East region. It is our pleasure to cite these articles and discuss it in our revised manuscript.
- The authors refer to their 2022 publication (ref. 10) when describing the inclusion of participants in the study. At the same time, the recruitment is also not fully described in that article; a link is given to another study by the authors (4). This is not convenient for readers, the authors should correct this section of the article.
Thank you for pointing this out. We corrected the manuscript. In the ‘Materials and Methods’ section we have added ‘Each year we received a pseudonymized list of Bialystok citizens from the Municipal Office in Bialystok. The dataset was limited to people aged 20-80 years old, and categories based on sex and 5-year intervals. We randomly sampled citizens from each subcategory separately, in such a number that allowed us to obtain a similar proportion distribution similar to that in the city’s population. There were any exclusions only some restrictions. Participants with an acute infectious disease or after surgery within the last six weeks were not examined and they were encouraged to come back to the study after this period. Moreover, during a coronavirus disease of 2019 (COVID-19) pandemic, reverse transcription-polymerase chain reaction (RT-PCR) was performed from nasopharyngeal swabs using the CFX96 Real-Time System (Bio-Rad) to exclude active COVID-19 infection. At the time of the study, there were exclusions for individual procedures, e.g. pregnancy for dual energy X-ray absorptiometry (DEXA) and diabetes for oral glucose tolerance test (OGTT).’
References:
- Li XL, Huang H, Lu Y, Stafford RS, Lima SM, Mota C, Shi X. Prediction of Multimorbidity in Brazil: Latest Fifth of a Century Population Study. JMIR Public Health Surveill. 2023 May 30;9:e44647. doi: 10.2196/44647.
- Tan MMC, Prina AM, Muniz-Terrera G, Mohan D, Ismail R, Assefa E, Keinert AÁM, Kassim Z, Allotey P, Reidpath D, Su TT. Prevalence of and factors associated with multimorbidity among 18 101 adults in the South East Asia Community Observatory Health and Demographic Surveillance System in Malaysia: a population-based, cross-sectional study of the MUTUAL consortium. BMJ Open. 2022 Dec 23;12(12):e068172. doi: 10.1136/bmjopen-2022-068172.
- Singh K, Alomari A, Lenjawi B. Prevalence of Multimorbidity in the Middle East: A Systematic Review of Observational Studies. Int J Environ Res Public Health. 2022 Dec 8;19(24):16502. doi: 10.3390/ijerph192416502.
- Chlabicz M, Jamiołkowski J, Paniczko M, Sowa P, Szpakowicz M, Łapińska M, Jurczuk N, Kondraciuk M, Ptaszyńska-Kopczyńska K, Raczkowski A, Szpakowicz A, Kamiński KA. ECG Indices Poorly Predict Left Ventricular Hypertrophy and Are Applicable Only in Individuals With Low Cardiovascular Risk. J Clin Med. 2020 May 6;9(5):1364. doi: 10.3390/jcm9051364.
All changes in the manuscript are in red.
Please accept my sincere thanks for the advice and work.
- Frolich, A.; Ghith, N.; Schiotz, M.; Jacobsen, R.; Stockmarr, A. Multimorbidity, healthcare utilization and socioeconomic status: A register-based study in Denmark. PLoS One 2019, 14, e0214183, doi:10.1371/journal.pone.0214183.
- Schafer, I.; von Leitner, E.C.; Schon, G.; Koller, D.; Hansen, H.; Kolonko, T.; Kaduszkiewicz, H.; Wegscheider, K.; Glaeske, G.; van den Bussche, H. Multimorbidity patterns in the elderly: a new approach of disease clustering identifies complex interrelations between chronic conditions. PLoS One 2010, 5, e15941, doi:10.1371/journal.pone.0015941.
- Dhungana, R.R.; Karki, K.B.; Bista, B.; Pandey, A.R.; Dhimal, M.; Maskey, M.K. Prevalence, pattern and determinants of chronic disease multimorbidity in Nepal: secondary analysis of a national survey. BMJ Open 2021, 11, e047665, doi:10.1136/bmjopen-2020-047665.
- Ma, X.; He, Y.; Xu, J. Urban-rural disparity in prevalence of multimorbidity in China: a cross-sectional nationally representative study. BMJ Open 2020, 10, e038404, doi:10.1136/bmjopen-2020-038404.
- Singh, K.; Alomari, A.; Lenjawi, B. Prevalence of Multimorbidity in the Middle East: A Systematic Review of Observational Studies. Int J Environ Res Public Health 2022, 19, doi:10.3390/ijerph192416502.
- Charlson, M.E.; Pompei, P.; Ales, K.L.; MacKenzie, C.R. A new method of classifying prognostic comorbidity in longitudinal studies: development and validation. J Chronic Dis 1987, 40, 373-383, doi:10.1016/0021-9681(87)90171-8.

Reviewer 2 Report
Thank you for your important study on multimorbidity in the urban population of Poland.
Your manuscript should be proofread once more for grammatical errors. Please find below some of my suggestions that may improve your study. Once these will be addressed, I am happy to support your work being published.
Title: After reading the paper I understood that the multimorbidities investigated were all physical. Please add this in the title. You should also add in the title that this study is for urban populations in Poland.
Abstract: Please proofread, there are grammatical errors. ‘limited data on multimorbidity’ – there are actually a number of studies conducted on multimorbidity, among different populations groups, i.e. older or young, with different patterns etc. I think it is an overstatement to say there is limited data so please rephrase such as: A number of studies have been conducted on multimorbidity, however….’. Your aim is clear but doesn’t mention where this study is conducted, please add Poland. Do same in your study title. The abstract has a lot of results stated, so it will be better to summarise them before ending the para with the healthcare service suggestion.
Introduction: Please proofread as there are a number of grammatical errors. Your first para are good and informative. Please add some more previous studies conducted on urban population with multimorbidity. If you can’t find any then please mention this in the introduction. You should add the previous studies on urban populations before stating the aim of your study in lines 62-65.
Methods: Was there an ethical approval provided for this survey? It will be good to mention this in the methods.
Can the authors add all the information on measurements (lines 79-134) in a table, to help the readers? It is a lot of information, and it will be clearer if the info is added in tables, as supplementary for example or main tables. Then you can summarise these in the methods text by excluding the exact measurement levels etc.
Interesting finding to see the differences between males and females. Have the authors compared them? If so could you add if the differences were statistically significant?
Your figures overall are very impressive. Could you add in the statistical part which software you used for them, as I am not aware if SPSS can create this. I find that Figure 5 is a bit more complicated to read as it has 4 categories. To my understanding student cases were very small, so could the authors have figure 5 only for the rest of the categories? This may make the figure more easy to read.
Table 4 seems to present the mean loss for all the population in the cohort and then the para below is presenting this by gender, educational and employment status, which are further added in supplementary tables. Can the authors summarise this para (228-242) in a way that the most significant results stand out?
Discussion: 249-250 the authors state: ‘High heterogeneity between studies in our meta-analysis implies that the prevalence of multimorbidity varies between studies’. Are the authors referring to another study of theirs, as the current one is not a metanalysis. Please clarify.
Authors have used several previous studies in the discussion to provide differences and similarities with their results. In the end of the discussion, they jump directly to the suggestion of healthcare service providers without adding any previous text on the utilisation of services for those with multimorbidities, mainly the ones clearly standing out in the current study. This will be helpful to have. If there is a word limit in this submission, the authors can minimise previous parts of the discussion and add further text on healthcare utilisation.
Conclusions: If discussion will be updated with some extra text on the healthcare utilisation, the conclusion can stay as it is.
Study limitations: I am not sure if this is related with the journal format but we usually have the study limitations within the main paper, i.e. after the discussion. If this is an option I suggest to place it there instead of this section.
Must be proofread for grammatical errors.
Author Response
Reviewer #2
Thank you for your important study on multimorbidity in the urban population of Poland.
Your manuscript should be proofread once more for grammatical errors. Please find below some of my suggestions that may improve your study. Once these will be addressed, I am happy to support your work being published.
Title: After reading the paper I understood that the multimorbidities investigated were all physical. Please add this in the title. You should also add in the title that this study is for urban populations in Poland.
Thank you for your suggestion. We corrected the title. The fact that physical illnesses were taken was highlighted in the abstract.
Abstract: Please proofread, there are grammatical errors. ‘limited data on multimorbidity’ – there are actually a number of studies conducted on multimorbidity, among different populations groups, i.e. older or young, with different patterns etc. I think it is an overstatement to say there is limited data so please rephrase such as: A number of studies have been conducted on multimorbidity, however….’. Your aim is clear but doesn’t mention where this study is conducted, please add Poland. Do same in your study title. The abstract has a lot of results stated, so it will be better to summarise them before ending the para with the healthcare service suggestion.
Thank you for your great comments. We changed the abstract as suggested.
Introduction: Please proofread as there are a number of grammatical errors. Your first para are good and informative. Please add some more previous studies conducted on urban population with multimorbidity. If you can’t find any then please mention this in the introduction. You should add the previous studies on urban populations before stating the aim of your study in lines 62-65.
We have made a linguistic correction.
In the ‘Introduction’ section we have added: ‘The prevalence of multimorbidity in the urban population is higher than in the rural population’
Methods: Was there an ethical approval provided for this survey? It will be good to mention this in the methods.
We've changed the place of the ethical statement right after the ‘Materials and Method’ section.
Can the authors add all the information on measurements (lines 79-134) in a table, to help the readers? It is a lot of information, and it will be clearer if the info is added in tables, as supplementary for example or main tables. Then you can summarise these in the methods text by excluding the exact measurement levels etc.
Thank you for your advice. We have prepared Table 1 (Table 1. Parameters, methods and equipment used in the study). This will be more convenient for the reader.
Interesting finding to see the differences between males and females. Have the authors compared them? If so could you add if the differences were statistically significant?
We performed additional analyses. All diseases were compared in terms of gender, education level and occupation. Additional columns with p-value were added to Table S1.
Your figures overall are very impressive. Could you add in the statistical part which software you used for them, as I am not aware if SPSS can create this. I find that Figure 5 is a bit more complicated to read as it has 4 categories. To my understanding student cases were very small, so could the authors have figure 5 only for the rest of the categories? This may make the figure more easy to read.
Thank you for your comment. We did not include this information. In the ‘Statistical analysis’ section we have added ‘The figures were prepared using the R version 4.2.2 software (www.R-project.org) and the ggplot2 library’.
Table 4 seems to present the mean loss for all the population in the cohort and then the para below is presenting this by gender, educational and employment status, which are further added in supplementary tables. Can the authors summarise this para (228-242) in a way that the most significant results stand out?
The indicated paragraph of the manuscript has been corrected:’ Based on patient-declared and newly diagnosed diseases, the average health loss among the urban population was 0.238. Migraine (12.6%), osteoarthritis (9.13%) and discopathy (8.99%) were the most significant factors in total health losses. The average health loss was 0.197 for men and 0.272 for women. The most significant relative role in total health loss among men established diabetes (11.6%), among women migraine (17.3%). In the subgroup with primary education, the average health loss was estimated at 0.361; in the subgroup with secondary education at 0.279 and in the subgroup with higher education at 0.194. The most important relative role in total health loss among those with primary education was migraine (12.8%), with secondary education osteoarthritis (10.1%), with higher education migraine (16.8%). In the students, the average health loss was estimated at 0.055; in the working individuals at 0.183, in the unemployed at 0.207, and in the pensioner at 0.491.’
Discussion: 249-250 the authors state: ‘High heterogeneity between studies in our meta-analysis implies that the prevalence of multimorbidity varies between studies’. Are the authors referring to another study of theirs, as the current one is not a metanalysis. Please clarify.
We refer to the meta-analysis of Chowdhury et al. We corrected the error.
Authors have used several previous studies in the discussion to provide differences and similarities with their results. In the end of the discussion, they jump directly to the suggestion of healthcare service providers without adding any previous text on the utilisation of services for those with multimorbidities, mainly the ones clearly standing out in the current study. This will be helpful to have. If there is a word limit in this submission, the authors can minimise previous parts of the discussion and add further text on healthcare utilisation.
Thank you for your comment. In the ‘Discussion’ section we have added ‘We identified a high prevalence of multimorbidity in an urban population in Poland, based on known and unknown conditions. This may be highly relevant for health care providers and may focus the search for unknown disease entities and hence multimorbidity in the relevant subpopulations. Our work has shown that more frequently unrecognised diseases were more common in the subgroup with primary education. Early detection and treatment of these diseases can result in lower health care costs, since a significant increase in the use of health care was shown for people with multimorbidity compared to people without chronic diseases’.
Conclusions: If discussion will be updated with some extra text on the healthcare utilisation, the conclusion can stay as it is.
The proposals have been revised: ‘Our work may help in the selection of appropriate screening tests based on age, sex and educational attainment in order to recognise both known and unknown conditions. Ultimately, it may have an impact on clinical practice, service delivery and study design.
Study limitations: I am not sure if this is related with the journal format but we usually have the study limitations within the main paper, i.e. after the discussion. If this is an option I suggest to place it there instead of this section.
We changed the place of this section as you recommend.
All changes in the manuscript are in red.
Please accept my sincere thanks for the advice and work.

Round 2
Reviewer 1 Report
The authors responded to my comments and corrected the text of the manuscript. I have no other comments.
No comments